# Architecture of the human G-protein-methylmalonyl-CoA mutase nanoassembly for B$_{12}$ delivery and repair

Romila Mascarenhas[1,2], Markus Ruetz[1,2], Harsha Gouda[1], Natalie Heitman[1], Madeline Yaw[1] & Ruma Banerjee [1] ✉

G-proteins function as molecular switches to power cofactor translocation and confer fidelity in metal trafficking. The G-protein, MMAA, together with MMAB, an adenosyltransferase, orchestrate cofactor delivery and repair of B$_{12}$-dependent human methylmalonyl-CoA mutase (MMUT). The mechanism by which the complex assembles and moves a >1300 Da cargo, or fails in disease, are poorly understood. Herein, we report the crystal structure of the human MMUT-MMAA nano-assembly, which reveals a dramatic 180° rotation of the B$_{12}$ domain, exposing it to solvent. The complex, stabilized by MMAA wedging between two MMUT domains, leads to ordering of the switch I and III loops, revealing the molecular basis of mutase-dependent GTPase activation. The structure explains the biochemical penalties incurred by methylmalonic aciduria-causing mutations that reside at the MMAA-MMUT interfaces we identify here.

Intracellular trafficking pathways shepherd rare, reactive, but essential metal and organic cofactors that undergird metabolism. With B$_{12}$, clinical genetics studies[1] had provided early insights into the multitude of handlers that shepherd and repair this cofactor, which is needed by just two mammalian enzymes: cytoplasmic methionine synthase and mitochondrial methylmalonyl-CoA mutase (MMUT)[2,3]. In both enzymes, the cobalt ion in B$_{12}$ (or cobalamin) cycles between different oxidation states and is susceptible to inactivation[4]. In methionine synthase, inactive cob(II)alamin is repaired in situ in the presence of an electron and a methyl group donor, to restore the active methylcobalamin form[5,6]. In MMUT, the inactive cob(II)alamin is physically off-loaded onto adenosyltransferase (MMAB) in a process powered by GTP hydrolysis and catalyzed by a second chaperone, MMAA (Fig. 1A)[7–9]. MMAB converts cob(II)alamin to 5′-deoxyadenosylcobalamin (AdoCbl) in the presence of ATP and an electron donor, and then re-loads the active cofactor form onto MMUT[7,9,10]. Structural insights into how such a gargantuan cofactor (1329 or 1579 Da for inactive/active forms) is loaded/off-loaded, and the inter-protein signals that orchestrate these processes, have been elusive. Mutations in each of the three mitochondrial B$_{12}$ trafficking proteins

are associated with hereditary methylmalonic aciduria, an inborn error of metabolism with a prevalence of ~1:90,000 births[1]. MMUT catalyzes the isomerization of methylmalonyl-CoA (M-CoA) funneled from the catabolic VOMIT (valine, odd-chain fatty acid, methionine, isoleucine and threonine) pathway reactions to succinyl-CoA, which enters metabolic mainstream[11]. MMUT deficiency leads to the anaplerotic insufficiency of TCA cycle intermediates that can potentially be restored by a membrane soluble form of α-ketoglutarate[12].

MMUT initiates substrate isomerization by homolytic cleavage of the cobalt-carbon bond in AdoCbl, generating cob(II)alamin and the 5′-deoxyadenosyl radical[13]. Inadvertent loss of the 5′-deoxyadenosine moiety from the MMUT active site leads to an inactive enzyme with cob(II)alamin bound (Fig. 1A)[14]. The structural basis for how apo- or inactive human MMUT signals to and recruits MMAA and MMAB for cofactor loading/off-loading is not known. Our understanding of how MMAA gates cofactor transfer to/from MMUT has emerged primarily from studies on the homologous proteins in *Methylobacterium extorquens*[7,10,14–17] and the *Cupriavidus metallidurans* fusion protein Icmf, comprising isobutyryl-CoA mutase and its G-protein chaperone[18,19]. While the

[1]Department of Biological Chemistry, University of Michigan, Ann Arbor, MI 48109, USA. [2]These authors contributed equally: Romila Mascarenhas, Markus Ruetz. ✉e-mail: rbanerje@umich.edu

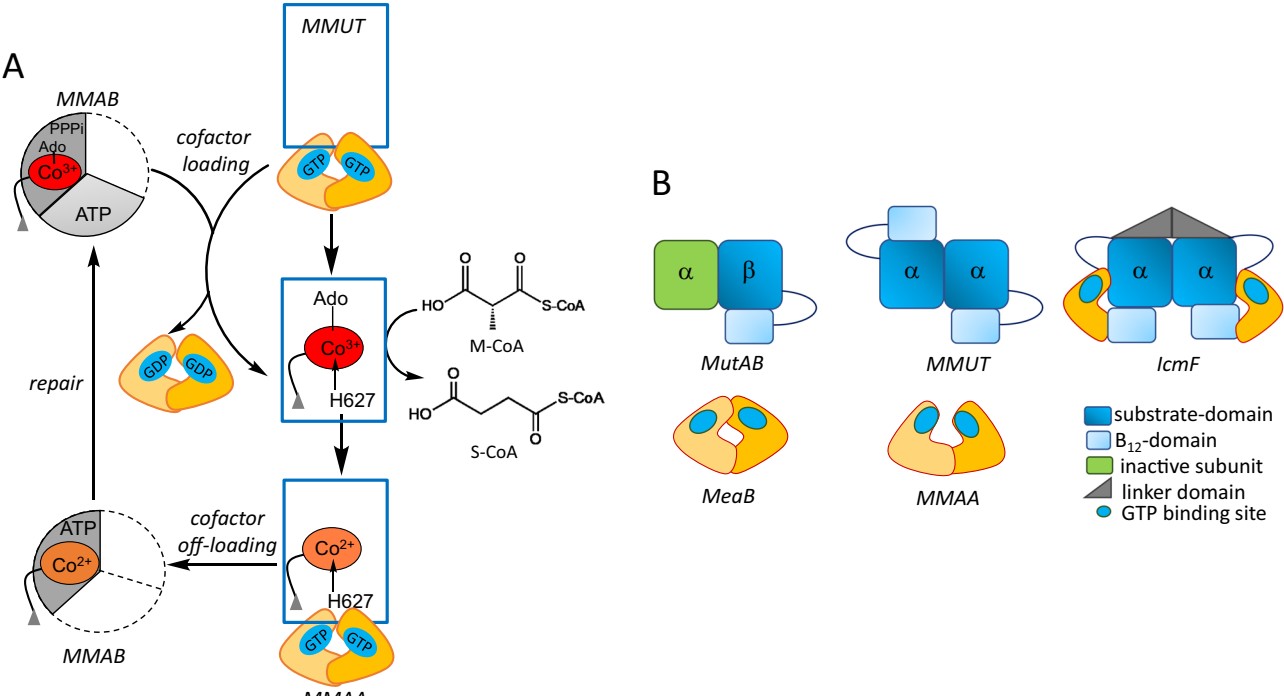

**Fig. 1 | Mitochondrial human B$_{12}$ trafficking proteins and comparison in their organization with bacterial homologs. A** Model for cofactor loading and off-loading in the mitochondrial B$_{12}$-trafficking pathway. After AdoCbl is loaded onto MMUT (blue) by MMAB (grey), MMUT catalyzes the isomerization of methymalonyl-CoA (M-CoA) to succinyl-CoA (S-CoA). The GAP function of MMUT enhances the GTPase activity of MMAA (yellow), which assists in the transfer of cofactor to/from MMUT. MMAB is an adenosyltransferase and converts cob(II) alamin to AdoCbl and loads it onto MMUT. **B** Cartoons showing the topological differences between bacterial MutAB, human MMUT, and IcmF.

bacterial and human orthologs of MMAA and MMUT share ~50% sequence identity, they exhibit large topological differences (Fig. 1B), which limit their utility as models of the human proteins and the biochemical penalties associated with their disease-causing variants.

The *M. extorquens* orthologs of MMAA (MeaB) and MMUT (MutAB) are tightly bound; their affinity is modulated by the G-nucleotide and cobalamin ligands bound to the respective proteins[16,17]. MeaB is a versatile chaperone that assists AdoCbl transfer from MMAB to MutAB, and cob(II)alamin transfer in the opposite direction, while also protecting the cofactor on MutAB against inactivation[14,16,17,20–22]. The canonical G-protein switch I and II loops afford nucleotide-responsive allosteric regulation of MeaB functions and suppress its intrinsic GTPase activity[20,22]. A third conformationally plastic switch III loop is important for the gating and editing functions of MeaB, which are corrupted by patient mutations that localize to this region[8,20]. Crystal structures of MeaB have captured the switch III loop in multiple nucleotide-sensitive poses[19,20]. Switch I and III loops are disordered in the structure of human MMAA, limiting insights[23].

Each IcmF monomer comprises an N-terminal B$_{12}$-binding Rossmann fold domain, a nucleotide-binding G-domain, a structured linker forming the dimer interface, and the C-terminal substrate-binding TIM barrel domain (Fig. 1B)[18]. While the G-domains in IcmF are structurally homologous to MeaB and MMAA, they are located at opposite ends, precluding dimer formation. The conserved nucleotide binding site is located at the interface between the substrate and G-domains.

MeaB and MMAA are homodimers with low intrinsic GTPase activity, which are activated in complex with the cognate mutase[17,24]. The human and bacterial G-proteins have distinct interfaces and their nucleotide binding sites reside on the same or opposite faces of the dimer, respectively at distances of 25 Å (MMAA) and 50 Å (MeaB). The oligomeric composition of the mutase is different between the bacterial (αβ, MutAB) and human (α$_2$) homologs, which translates into

architectural differences in their complexes with the corresponding G-proteins (Fig. 1B). The bacterial proteins exist in a 2:1 complex of MutAB:MeaB[22]. A recent structure of the complex between the B$_{12}$ domain of MutAB and MeaB in the presence of GMPPCP, captured a large conformational change, leading to the reorganization of the MeaB switch III loop and organization of the GTPase site[25]. In contrast to the *M. extorquens* assembly, the human proteins exist as a variety of free and equilibrating oligomeric complexes varying from linear to annular with a predominant stoichiometry of 1:1 MMUT:MMAA (designated $M_nC_n$ where n = 2 as in the $M_2C_2$ complex)[8]. Patient mutations in the switch III region of MMAA impact the distribution between these oligomeric forms[8].

Herein, we report the crystal structure of the human $M_2C_2$ complex with coenzyme A (CoA) and GDP, which reveals dramatic conformational changes that occur as a prelude to cofactor translocation. MMAA stabilizes a 180° rotation of the B$_{12}$ in relation to the substrate domain in MMUT by wedging between the two and undergoes ordering of its switch I and III loops. In this exploded conformation, the B$_{12}$ domain moves from a sequestered to a solvent-exposed pose, revealing how it is primed for cofactor loading/off-loading and explaining the mechanistic basis of pathogenic mutations that localize at the interfaces formed in the $M_2C_2$ nano-assembly.

## Results

The crystal structure of the human $M_2C_2$ complex with CoA and GDP•Mg$^{2+}$ was solved by molecular replacement at a resolution of 2.8 Å (Table S1). Each molecule in the asymmetric unit contains two dimers each of MMUT and MMAA (Fig. 2A), forming an annular structure as seen previously by negative staining[8]. Each monomer in the MMUT α$_2$ dimer comprises an N-terminal TIM barrel domain that binds substrate and a C-terminal Rossman fold domain that binds B$_{12}$[23]. The substrate domains form the interface in the head-to-toe dimer arrangement, positioning the two B$_{12}$ domains at opposite ends

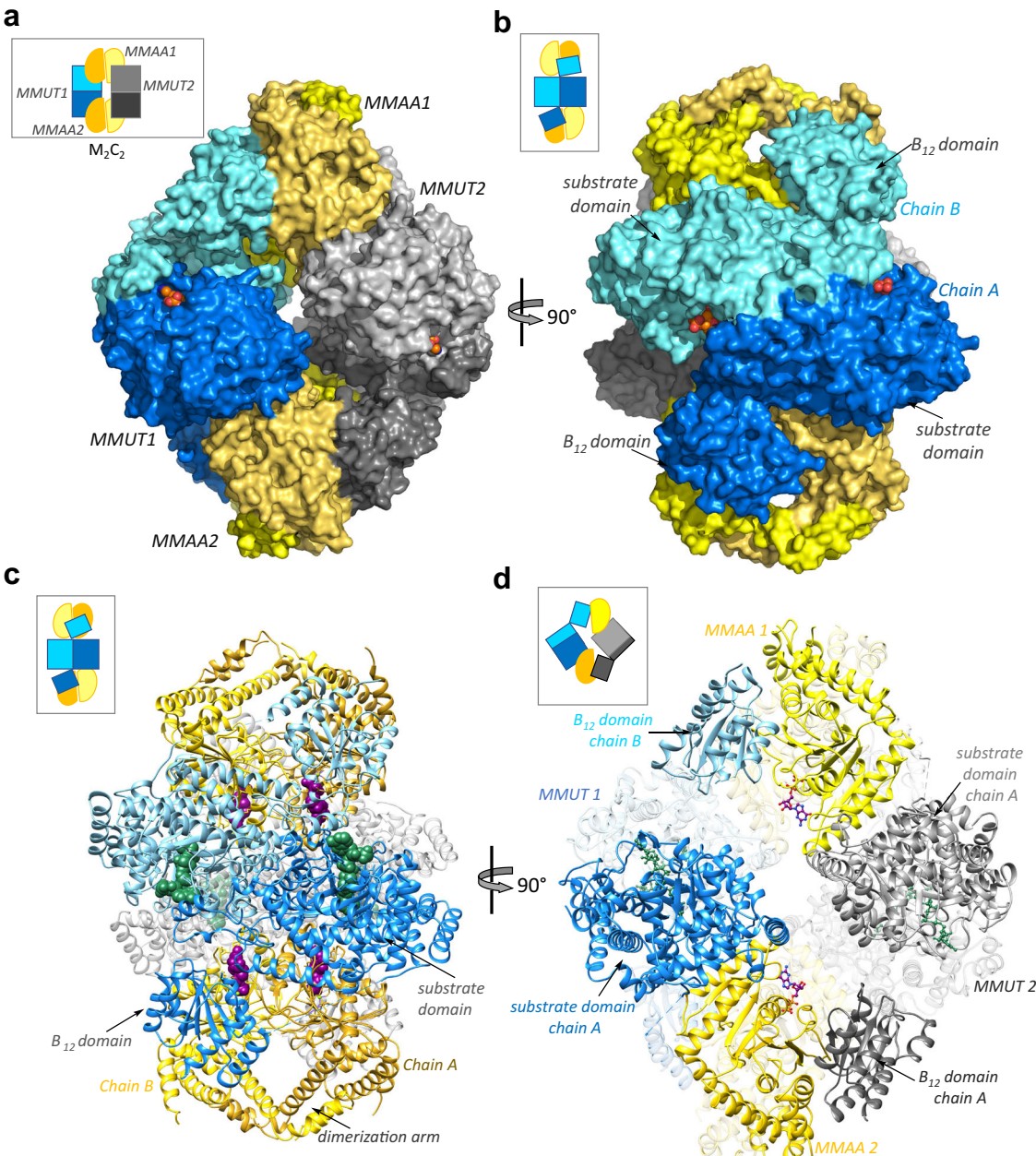

**Fig. 2 | Structure of the human $M_2C_2$ complex. A** Surface representation of the $M_2C_2$ structure. MMUT1 and 2 are shown in blue and grey, respectively and the dark and light shades of MMAA (yellow) represent the monomeric chains in each $\alpha_2$ dimer. **B** A 90° rotation shows that each MMUT chain (dark and light blue) interacts with one MMAA dimer. **C, D** Ribbon representation of the $M_2C_2$ complex. GDP (purple) and CoA (green) are shown as spheres. **D** One chain in MMAA2 (gold) simultaneously interacts with the substrate-domain in MMUT1 (dark blue) and the $B_{12}$-domain in MMUT2 (dark grey). GDP (purple) and CoA (green) are shown as sticks.

(Fig. 2B). The substrate domain of an MMUT monomer forms an interface with one chain of MMAA, while the $B_{12}$ domain interacts with a second MMAA chain (Fig. 2C). Conversely, one chain in each MMAA $\alpha_2$ dimer simultaneously interacts with the substrate domain from one MMUT and the $B_{12}$ domain from the second MMUT, leading to the annular form (Fig. 2D).

## MMUT in the $M_2C_2$ complex

The $B_{12}$ domain is rotated ~180° in the $M_2C_2$ complex versus in MMUT, which is evident from overlays of MMUT in the $M_2C_2$ complex on apo-MMUT (PDB code 3BIC), MMUT•AdoCbl (PDB code 2XIJ) or MMUT•CoA•AdoCbl (PDB: 2XIQ) (Cα RMSD across all pairs = 17 Å) (Fig. 3A, Supplementary Movie 1). MMAA is wedged between the two domains,

stabilizing the exploded MMUT conformation (Supplementary Fig. 1A) that is secured by multiple interactions between them. The swinging out of the $B_{12}$ domain is enabled by a flexible interconnecting belt (~100 residues), which is partially unresolved in the $M_2C_2$ structure (between 580–595), suggesting mobility. In free MMUT, AdoCbl is nestled between the stacked $B_{12}$ and substrate domains and shielded from solvent by an ordered belt (Fig. 3A). In the $M_2C_2$ complex, this interface is disrupted by the movement of the $B_{12}$ domain, which exposes it to solvent. The MMUT loop (residues 622–629) that carries the conserved $B_{12}$ ligand (His-627), is partially disordered in the "unstacked" conformation. The CoA threads through a channel in the substrate domain inducing it to close in on itself, as seen previously in the structure of MMUT bound to a substrate analog[23].

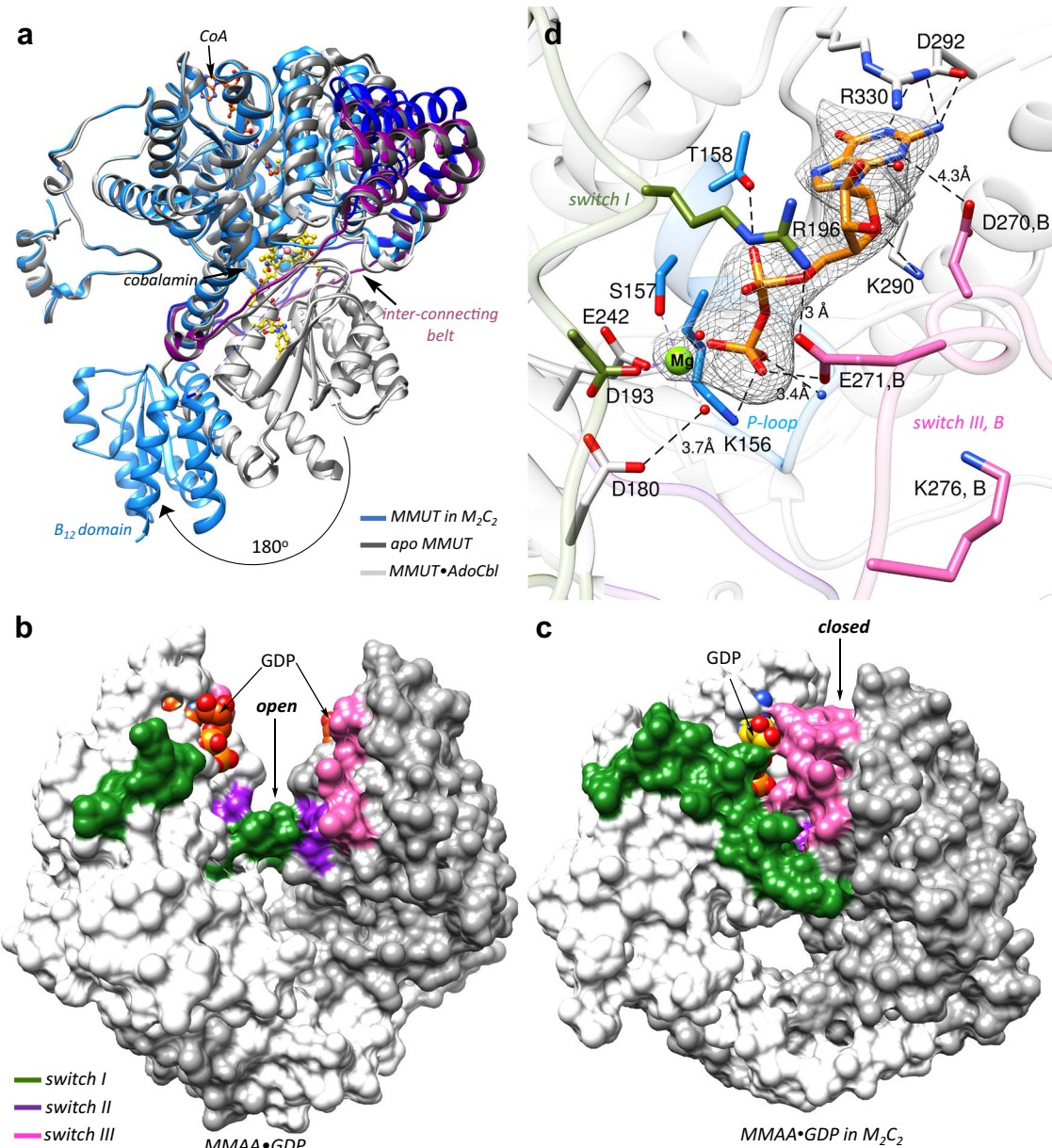

**Fig. 3 | Conformational changes in MMUT and MMAA in the $M_2C_2$ complex.**
**A** Overlay of MMUT in the $M_2C_2$ complex (blue) on free apo-MMUT (PDB: 3BIC)
(dark grey) or with CoA (orange) and AdoCbl (maroon) bound (PDB:2XIQ) (light
grey) shows that the $B_{12}$ domain is swung out by ~180° and is solvent exposed. In
MMUT•AdoCbl an interconnecting belt (purple) wraps around the active site and
protects the cofactor from solvent. In $M_2C_2$ the interconnecting belt is partially
disordered (dark blue) **B**, **C** Comparison of free MMAA (PDB: 2WWW) (B) and
MMAA in $M_2C_2$ (C), shows that it undergoes a large shift from an open to a closed

conformation in the $M_2C_2$ complex, which leads to ordering of the switch I[62] and III
(pink) loops. **D** The MMAA active site in $M_2C_2$ shows that GDP (orange sticks)
binding is stabilized by multiple hydrogen bonds as described in the text. Asp-270
and Glu-271 from the switch III loop of chain B (pink) also stabilize GDP binding.
$Mg^{2+}$ is coordinated by four oxygens donated by Ser-157, Glu-242, Asp-193, and
oxygen from the β-phosphate of GDP. mFo-DFc omit map of GDP and $Mg^{2+}$ at 2 σ is
shown as a grey mesh.

## MMAA in the $M_2C_2$ complex

Each MMAA monomer in the $\alpha_2$ dimer comprises the following sig-
nature motifs: switch I, II, and III for signal transduction, a P-loop, a
base specificity loop, and a C-terminal dimerization arm. A structural
comparison between free (PDB: 2WWW) and $M_2C_2$-bound (Cα
RMSD = 3.2 Å) MMAA reveals that one chain undergoes a large inward
rotation by ~60° towards the dimer interface that is driven by an ~23 Å
movement of α-helix7 (residues 296 to 311) in the complex (Supple-
mentary Fig. 1B, C, Supplementary Movie 2). While the distance
between the GDP sites (ribose O'3) in the dimer is unchanged (i.e.,
27–28 Å), GDP in the rotated chain moves by 14 Å relative to its position
in free MMAA, positioning it closer to the dimer interface and away

from solvent (Supplementary Fig. 1C). This open-to-closed motion is
more clearly visualized in the surface display of the MMAA structures
(Fig. 3B, C). The hinge action of the dimerization arm enables the
conformational transition from the open to the closed state. A similar
albeit larger ~180° rotation of one subunit of MeaB relative to the
other, which buries a previously exposed nucleotide binding site at the
dimer interface, was seen in the recent structure of the MeaB-MutAB
$B_{12}$ domain complex (Supplementary Fig. 1D) and shortened the dis-
tance between the nucleotides from 45 Å to 18 Å in the complex[25]. The
partially resolved switch I and III loops in free MMAA[23] are resolved in
the complex with multiple interactions serving to staple their con-
formations (Supplementary Fig. 1E, F). Clear electron density is seen

for the switch III region in two out of four MMAA chains in the $M_2C_2$ complex, with the switch III loop from chain B interacting with GDP in chain A (Supplementary Fig. 1E). However, the modest side chain electron density of switch III residues, limits accurate modeling of its interaction with GDP. Our model places the side chains of Asp-270 within 4.3 Å to the guanine moiety and Glu-271 within hydrogen bonding distance to the β-phosphate moiety, respectively of GDP (Fig. 3D, Supplementary Fig. 1E). Additionally, switch II loops from adjacent subunits interact via the backbone atoms of Val-246 (Supplementary Fig. 1E). The switch III loop was previously captured in three solvent-exposed poses in MeaB[15,20]. In the MeaB-MutAB $B_{12}$ domain complex, switch III from subunit B makes key contacts with GMPPCP in subunit A via residues that are conserved in MMAA[25] (Supplementary Fig. 2A). Additionally, Lys-276 in MMAA is positioned to serve a similar role as Lys-188 in MeaB, in coordinating to the γ-phosphate in GTP (Fig. 3D, S2A).

GDP is anchored via a salt bridge with Asp-292 and hydrogen bonds with Lys-290, Arg-330, Thr-158, and Lys-156, and with backbone interactions with P-loop residues (154–155) as seen previously in the structure of free MMAA bound to GDP[23] (Fig. 3D). The GDP site is however, more ordered in the structure of $M_2C_2$ with $Mg^{2+}$ coordinated by three oxygens from the sidechains of Ser-157, Glu-242, Asp-193 and oxygen from the β-phosphate group of GDP, and two additional water molecules complete the coordination sphere. While the residue corresponding to Asp-180 in MeaB (Asp-92) was predicted to serve as a general base for activating water, its mutation to alanine did not impair the intrinsic GTPase activity[22]. In the $M_2C_2$ complex Arg-196 from switch I forms a salt bridge with Glu-271 (Fig. 3D). Similarly, in the MeaB-MutAB $B_{12}$ complex, the analogous residue (Arg-108) forms a salt bridge with Asp-182 connecting switch I in chain A to switch III in chain B (Supplementary Fig. 2A). The GDP site in the $M_2C_2$ complex reveals how the product is bound and that conformational changes in switch III would be needed to accommodate the additional phosphate in the substrate, GTP.

## MMUT•MMAA interface

In the $M_2C_2$ complex, the $B_{12}$ domain of MMUT is swapped out by chain A of MMAA, creating a new interface (Fig. 4A). An MMUT loop in the substrate domain (spanning residues 227–234), undergoes a large rotation that orients Arg-228 and Tyr-231 for interactions with Asp-344 and Asp-340, respectively on MMAA (Supplementary Fig. 3A). The side chain of Arg-228 is mobile and forms a hydrogen bond with the side chain of the corrin ring in the free MMUT•AdoCbl structure or is rotated away to form a salt bridge with the substrate analog in the MMUT•AdoCbl•malonyl-CoA structure[23]. Additional hydrogen bonds that zip this new interface include the backbone carbonyls of Glu-360, Ala-393, and Asp-498 on MMUT interacting with the side chains of Arg-300 and Arg-301 on MMAA, and the side chains of Asp-156 and Gln-476 on MMUT interacting with Ser-336 and Ser-308, respectively (Fig. 4B, Supplementary Fig. 4A).

A new interface is also created between the $B_{12}$ domain and MMAA chain B (Fig. 4A). Switch I residues (182–198), which are unresolved in free MMAA, are ordered in a short β-sheet (189–191), which hydrogen bonds with the terminal β-strand in the $B_{12}$ domain, creating an extended β-sheet connection between the two proteins (Fig. 4C). A similar β-sheet extension from the $B_{12}$- to the G-domain was observed in IcmF and in the MeaB-MutAB $B_{12}$ domain complex (Supplementary Fig. 3B)[18,25]. Interestingly, the switch I loop, which links MMAA to the $B_{12}$ domain, also links to switch III in the adjacent MMAA chain (via Arg-196) (Fig. 3D.) Although the $B_{12}$ domain interacts primarily with chain B, it also connects via a salt bridge between Asp-640 and Arg-277 to the switch III of chain A (Fig. 4C), analogous to the interaction between Asp-609 in the MutAB $B_{12}$ domain and Lys-189 in MeaB[25]. The $B_{12}$-MMAA interface is further stabilized by salt bridges between Arg-98 and Asp-663 and Arg-202 and Gln-652 and a hydrogen bond

between the backbone atoms of Gly-187 and the side chain of Arg-614 (Fig. 4C, Supplementary Fig. 4B). Additionally, the backbone carbonyl of Pro-210 hydrogen bonds with Arg-616 (Fig. 4D, Supplementary Fig. 4C). The side chains of Tyr-207 and Arg-209 form hydrogen bonds with the backbone of Asp-663. Arg-214 forms a salt bridge with Asp-644, and Arg-202 and Thr-198 hydrogen bonds with the backbone of Phe-651 and the side chain of Gln-652, respectively.

## Patient mutations at MMUT-MMAA interface impair function

The biochemical penalties associated with methylmalonic aciduria causing variants[26–30] that localize at the interfaces between MMAA and the MMUT substrate (R228Q) and $B_{12}$ (R616C and R694W) domains were assessed as a test of the biological relevance of the $M_2C_2$ structure (Supplementary Fig. 5A). The GTPase activating protein (GAP) activity of MMUT serves as a proxy for its complex formation with MMAA[8]. The intrinsic GTPase activity of MMAA ($0.06 \pm 0.02$ min$^{-1}$), which increases 36-fold ($2.1 \pm 0.2$ min$^{-1}$) in the presence of wild-type MMUT, is attenuated 7- (R616C) and 12-fold (R228Q and R694W) in the variants (Supplementary Table 2). The gating function of MMAA during AdoCbl transfer from MMAB to wild-type MMUT in the presence of GMPPCP was used as an additional test of complex formation (Supplementary Fig. 6A). Complete AdoCbl transfer to R616C MMUT but <10% transfer to R228Q and R694W MMUT was observed, indicating full or partial loss of the GTPase gating function (Supplementary Fig. 6B-E). While AdoCbl-loading from the solution was variously impacted by the mutations (Supplementary Fig. 7) the catalytic activity of R228Q was undetectable. The activities of the other variants, R616C ($71 \pm 4$ μmoles min$^{-1}$ mg$^{-1}$) and R694W ($73 \pm 1$ μmoles min$^{-1}$ mg$^{-1}$) were comparable to wild-type MMUT ($89 \pm 6$ μmoles min$^{-1}$ mg$^{-1}$) (Supplementary Table 2). Cob(II)alamin off-loading from MMUT to MMAB was variously impacted by the mutations; R228Q was unaffected, R694W was partially impaired and R616C was completely impaired (Supplementary Fig. 8). Finally, we tested the impact of simultaneously severing the links between MMAA and the substrate and $B_{12}$ domains of MMUT by engineering the R228Q/R616C double mutant. GAP activation, MMAA-gated AdoCbl loading, and cob(II)alamin off-loading functions were all lost (Supplementary Figs. 6, 8). In contrast, AdoCbl binding from the solution was unaffected (Supplementary Fig. 7), although catalytic activity was undetectable (Supplementary Table 2).

We also characterized two MMAA patient mutations, R98G[31] and R209S[32], which reside at the interface with the $B_{12}$ domain in the $M_2C_2$ complex (Supplementary Fig. 5B). While neither mutation affects the intrinsic GTPase activity, GAP activation by MMUT is either enhanced 2-fold (R98G) or lost (R209S) (Supplementary Table 2). Both mutations impaired nucleotide-gated transfer of AdoCbl from MMAB to MMUT, which was observed even in the presence of GMPPCP (Supplementary Fig. 6) and were unable to power cob(II)alamin off-loading from MMUT for repair (Supplementary Fig. 8). These data are consistent with these disrupting mutations disrupting $M_2C_2$ complex formation.

## Discussion

G-proteins play important roles in metal homeostasis pathways by mechanisms that are generally poorly understood. Structural insights into the mitochondrial G-protein, MMAA, have been limited by the variable stability and/or oligomerization state of its complex with MMUT, which is influenced by multiple ligands that bind to each protein. In this study, we report the dramatic conformational changes that accompany the $M_2C_2$ nano-assembly, exposing the $B_{12}$ domain as a prelude to cofactor translocation (Fig. 5).

In all previous structures of MMUT and its bacterial orthologs, the $B_{12}$ and substrate domains are stacked on each other, forming a solvent-protected active site[23,33,34]. The apo- and inactive forms of MMUT (with cob(II)alamin bound) exhibit a higher affinity for MMAA and the $M_2C_2$ structure explains how its assembly harnesses the GTPase activity to make the $B_{12}$ domain accessible for cofactor

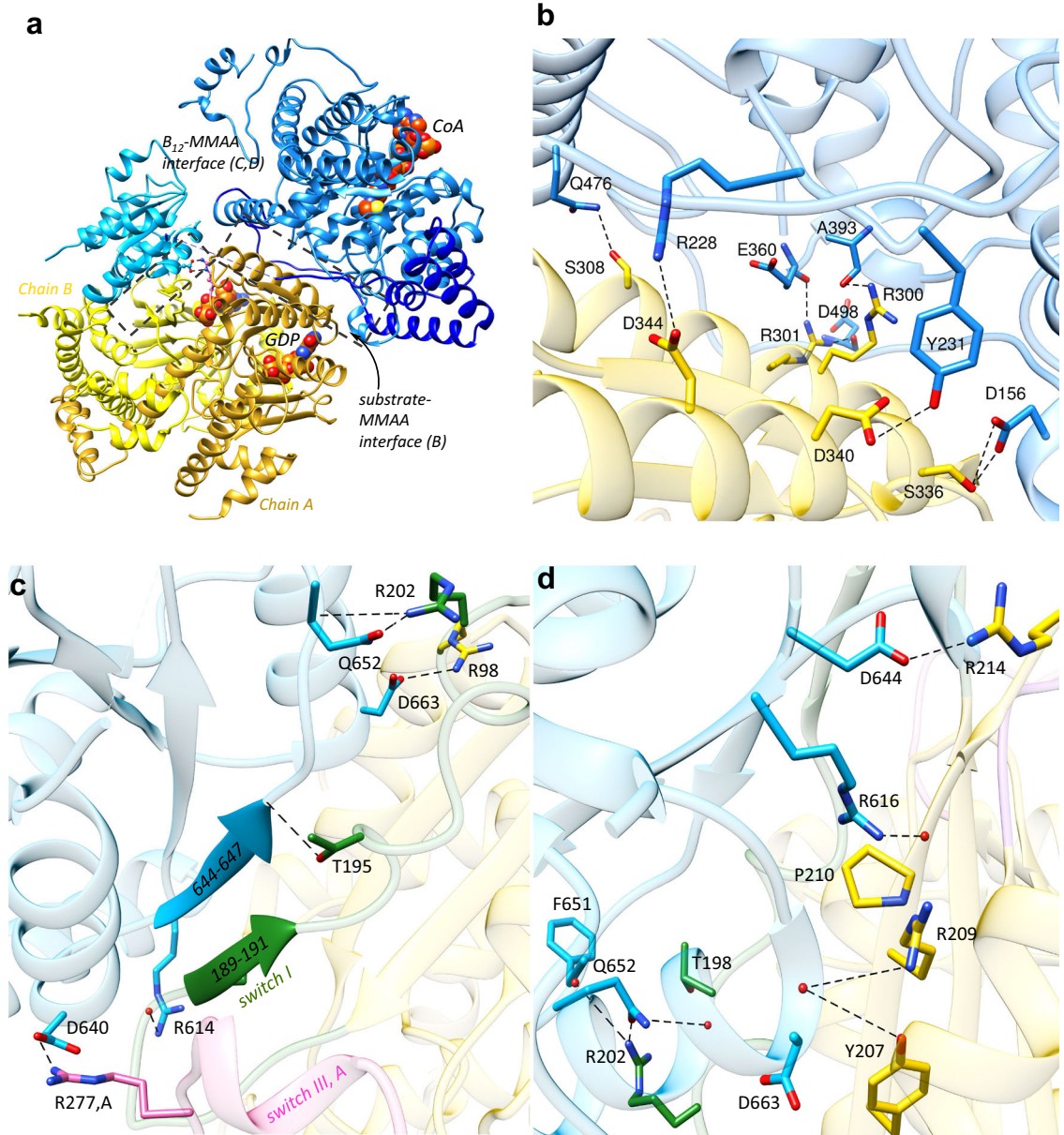

**Fig. 4 | MMUT-MMAA interfaces. A** Each MMUT subunit comprising the substrate- (blue) and $B_{12}$ (cyan) domains and the inter-connecting belt (dark blue) in $M_2C_2$ interacts with an MMAA dimer (yellow ribbons). CoA (orange) and GDP (orange) are shown as spheres. Boxes (dashed lines) outline two MMUT-MMAA interfaces. **B** Close-up of the MMUT substrate domain (blue) and MMAA (chain A, yellow) interface. Black dashed lines represent interactions between residues on MMUT and MMAA (chain A) as described in the text. **C, D** Close-up of the interface between the $B_{12}$ domain and MMAA (chain B). **C** Switch I on MMAA (green) is ordered, forming a short β-strand and hydrogen bonds with the terminal β-strand in the $B_{12}$ domain, resulting in an extended β-sheet interaction between the two proteins. Salt bridge and hydrogen bond interactions are shown with dashed lines. **D** Additional interactions (dashed black line) between MMAA and MMUT residues stabilize the $B_{12}$ domain-MMAA interface as described in the text.

transfer. Apo-MMUT can be reconstituted with free AdoCbl in vitro, suggesting that the unstacked conformation of MMUT is available even in the absence of MMAA (Fig. 5). However, this path is unlikely to be physiologically relevant since the cellular concentration of free AdoCbl is negligible[35]. MMAA wedges physically between the two MMUT domains and stabilizes a heretofore unseen conformation in which the $B_{12}$-domain is swung out by 180° degrees relative to the substrate domain, exposing the highly conserved His-627 cobalamin ligand to solvent. Mutation of the corresponding histidine ligand in MutAB leads to AdoCbl loading failure, revealing a critical role for the thermodynamically favored increase in coordination number (from 5- to 6-coordinate) for cofactor translocation from MMAB to MMUT[7]. Crystallographic snapshots of the *Mycobacterium tuberculosis* MMAB

ortholog captured multiple poses that provide clues into how AdoCbl, but not cob(II)alamin, is selectively translocated to the mutase[36].

Once loaded with AdoCbl, the affinity of holo-MMUT for MMAA is significantly weakened[8], leading us to propose that the catalytically active form of the mutase functions as a stand-alone dimer (Fig. 5). Inactivation of MMUT with resultant cob(II)alamin accumulation, necessitates cofactor repair with concomitant protection against hyperoxidation to the aquo-cob(III)alamin state, which represents a dead-end complex. We have recently discovered bivalent mimicry by ADP, an abundant metabolite, induces a conformational changes that protects cob(II)alamin from over-oxidation[37]. Cob(II)alamin off-loading to MMAB from this state is promoted by CoA (or M-CoA) binding to MMUT and by MMAA-dependent GTP hydrolysis (Fig. 5)[37].

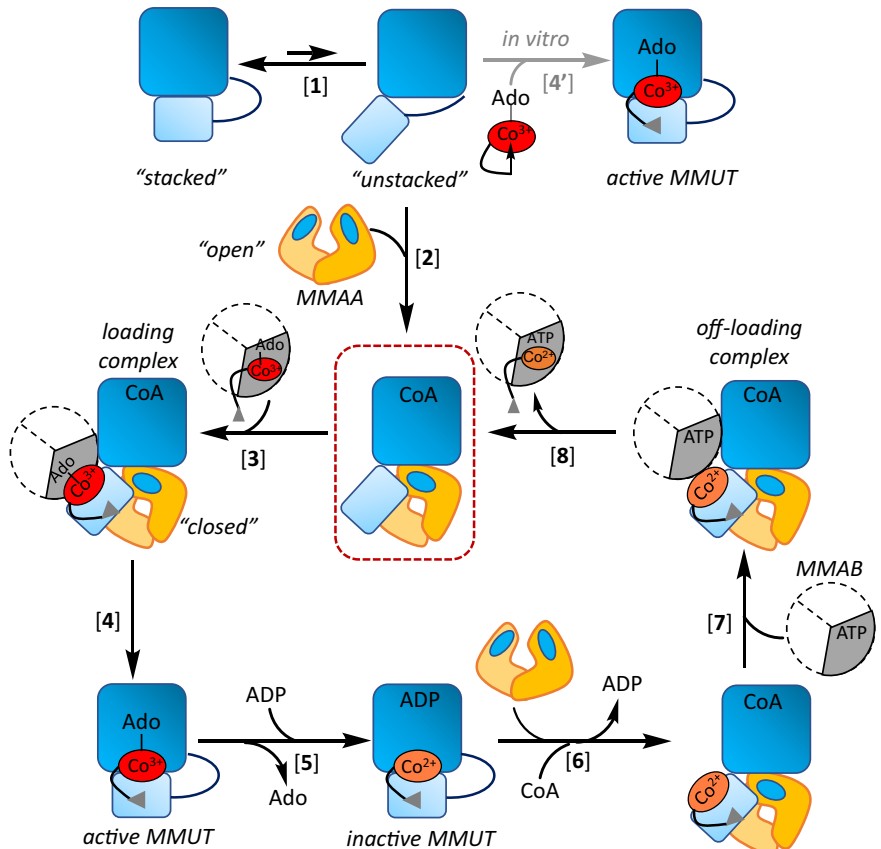

**Fig. 5 | Model of assemblies involved in B$_{12}$ loading and repair.** Free MMUT (light and dark blue are the B$_{12}$ and substrate domains) exists in the "stacked" and "unstacked" conformations with the equilibrium favoring the former [1]. MMAA (yellow) stabilizes the unstacked conformation by wedging between the substrate and B$_{12}$ domains and itself transitions from an "open" to "closed" state in M$_2$C$_2$, crystallized in this study (red box) [2]. MMAB transfers AdoCbl to M$_2$C$_2$ [3], leading to catalytically active holo-MMUT [4]. In vitro, free AdoCbl can bind to MMUT [4']. Inactivation of MMUT due to loss of Ado (deoxyadenosine) followed by ADP binding, stabilizes cob(II)alamin against hyperoxidation [5] and leads to recruitment of the repair complex [6] and [7]. Following cob(II)alamin off-loading and conversion to AdoCbl by MMAB, the active cofactor is reloaded onto MMUT [8].

It is presently unclear, however, how inactive MMUT (with cob(II)alamin and CoA) is distinguished from the active enzyme (with AdoCbl and M-CoA) to recruit the repair system (MMAB and MMAA).

Free MMAA exhibits an "open" conformation in which the nucleotide-binding site is only partially ordered, explaining its low intrinsic GTPase activity[23]. In the M$_2$C$_2$ complex, MMAA is in a "closed" conformation, which brings its active site into the register, explaining the GAP function of MMUT (Fig. 3C, D). Each switch III domain in the MMAA dimer, previously shown to be important for its GTPase activity[8,20], completes the nucleotide-binding site in the opposite monomer in the M$_2$C$_2$ complex. A similar switch III crossover strategy for building the active site was seen in the recent MeaB-MutAB B$_{12}$ domain structure[25]. Comparable switch III interactions are, however, absent in IcmF where the two G-domains are at opposite ends of the protein (Supplementary Fig. 2B)[18]. G-domain dimerization would require the linear association of two IcmF dimers. Formation of an active nucleotide binding site at the dimer interface was also observed in HypB, a G-protein involved in nickel trafficking, where residues from one chain contribute to nucleotide binding in the adjacent chain[38]. Interestingly, the interprotein β-sheet extension between switch I and the B$_{12}$ domain is a structural innovation that recurs in IcmF[18], MeaB-MutAB B$_{12}$ domain[25], and in the human M$_2$C$_2$ complex (Supplementary Fig. 3B). This motif confers rigidity to the interface and provides a direct conduit for signal transmission from the nucleotide to the B$_{12}$ site.

The unstacked conformation provides structural insights into how MMAB might access the B$_{12}$-binding site in the M$_2$C$_2$ complex and

the role of MMAA in facilitating this process (Fig. 5). We speculate that the GDP and CoA bound M$_2$C$_2$ complex captured in this study, could also serve a regulatory role for sequestering MMUT and prioritizing B$_{12}$ for the cytoplasmic branch[39]. In the liver, ~25% of MMUT is B$_{12}$ loaded, whereas cytoplasmic methionine synthase exists predominantly in the holo-form[40,41]. MMAA is predicted to be predominantly GDP-loaded based on the intracellular concentrations of GDP (160 μM) and GTP (~470 μM) relative to their respective $K_D$ values for MMAA (1.1 μM (GDP) and 740 μM (GTP))[8] Apo-MMUT is likely to predominantly CoA bound based on the $K_D$ (113 μM) versus the mitochondrial concentration of CoA (2–5 mM)[37] Thus, the M$_2$C$_2$ complex captured here could represent a holding structure that would be primed for loading upon exchange of GDP with GTP and availability of AdoCbl-bound MMAB.

The dramatic stacked to unstacked conformational change could be a common strategy used by B$_{12}$-dependent enzymes, particularly those that bind the cofactor in the base-off conformation. Ligand-triggered conformational changes have been reported in bacterial methionine synthase, although the mechanism of cofactor loading awaits elucidation[42]. In *Aquincola tertiaricarbonis*, a MeaB like gene is in the same operon as the large subunit of cobalamin-dependent hydroxyisobutyryl-CoA mutase[43]. Although the substrate (large) and B$_{12}$ (small) subunits are on separate polypeptides, the mechanism of B$_{12}$ loading onto this mutase could represent a variation on the same theme as in MMUT. On the other hand, AdoCbl-dependent diol dehydratase[44], glycerol dehydratase[45], ethanolamine ammonia lyase[46] and glutamate mutase[47] use an ATP-dependent reactivase to repair the inactive cofactor. The exchange of ADP with ATP releases the activase

and frees up the apo enzyme for AdoCbl loading[48]. A docking model of diol dehydratase with its reactivase suggests that the substrate domain must tilt relative to the $B_{12}$ domain to avoid steric clashes in the complex[49].

In summary, the structure of the $M_2C_2$ nanoassembly provides insights into a common intermediate in the bidirectional movement of the $B_{12}$ cofactor between human MMUT and MMAB (Fig. 5). The dramatic change in solvent access of the $B_{12}$-domain suggests that it serves as a platform for recruiting MMAB. MMAA also undergoes large conformational changes that complete its active site architecture and explains the molecular basis of the GAP activity of MMUT. Patient mutations at the MMAA-MMUT interfaces impact complex formation and highlight the importance of protein dynamics in translocating a large cofactor.

## Methods

### Materials
Adenosine 5′–triphosphate disodium salt hydrate (ATP) (Cat. # A2383), Coenzyme A disodium salt (CoA) (Cat. # C3144), guanosine 5′–triphosphate sodium salt hydrate (GTP) (Cat. # G8877), guanosine 5′–diphosphate sodium salt (GDP) (Cat. # G7127), β,γ-methyleneguanosine 5′-triphosphate sodium salt (GMPPCP) (Cat. # M3509), and 5′–deoxyadenosylcobalamin (AdoCbl) (Cat. # C0884) are from Sigma-Aldrich. Isopropyl β-D-1-thiogalactopyranoside (IPTG) (Cat. # I2481C) and Tris (2-carboxyethyl) phosphine (TCEP) (Cat. # TCEP) are from Gold Biotechnology. Ni(II)-NTA resin (Cat. # 30210) was from Qiagen. Primers were purchased from Integrated DNA Technologies. Cob(II) alamin was prepared by photolysis of AdoCbl as described previously[50].

### Expression and purification of wild-type and mutant MMUT
The following forward primer sequences were used for generating the R228Q, R616C, R694W, and R228Q/R616C MMUT mutants using the wild-type MMUT expression clone in a pET-28b vector with a C-terminal TEV cleavable His-tag. The reverse primer had a complementary sequence.

R228Q:5′GATATTCTGAAAGAATTTATGGTTCAAAACACCTACAT CTTCCCGCCGGAACCG-3′

R616C: 5′CGTCGCCCGTGCCTGCTGGTTGCCAA3′

R694W: 5′GAACTGAACAGCCTGGGCTGGCCGGATATTCTGGTC ATGTGT3′

The mutations were verified by Sanger nucleotide sequence analysis (Eurofins). Recombinant wild-type MMUT and MMUT variants were expressed as reported earlier[51]. In brief, *E.coli* BL21 (DE3) (EMD Millipore) was grown in Terrific Broth at 37 °C until $OD_{600nm}$ reached 1.5–1.8. Then, 3% (v/v) dimethyl sulfoxide (DMSO) was added, and the temperature was reduced to 16 °C (wild-type MMUT, R616C, and R694W) or 12 °C (R228Q and R228Q/R616C). After 40 min, 100 μM IPTG was added, and the culture was grown for 22–24 h. The cells were collected by centrifugation at $5500 \times g$ and stored at −80 °C. Wild-type and mutant MMUT were purified as described previously[51]. The cell pellet was resuspended in lysis buffer (50 mM Tris pH 8.0, 500 mM NaCl, 20 mM imidazole, 5% glycerol, 5 mL/g of wet cell pellet) in the presence of 100 μM phenylmethylsulfonyl fluoride and 1 tablet EDTA-free cOmplete™ Protease Inhibitor Cocktail (Roche). The suspension was sonicated, and cell debris was removed by centrifugation at $38,500 \times g$. The supernatant was loaded onto a Ni-NTA column, and washed with 150 mL lysis buffer. MMUT was eluted from the column with a liner imidazole gradient (20–200 mM). Fractions containing MMUT were pooled and concentrated to 20–30 mL using an Amicon 50 kDa centrifugal device (EMD Millipore). His-tagged TEV protease was added (0.02 mg/mg of protein) and dialyzed overnight against 50 mM Tris pH 8.0, 300 mM NaCl, 1 mM DTT 0.5 mM EDTA, and 5% glycerol. The TEV-treated solution was loaded onto a Ni-NTA column and the cleaved MMUT, present in the flowthrough, was concentrated

down to ~20 mL using an Amicon 50 kDa centrifugal device. After overnight dialysis against 50 mM HEPES pH 7.3, 25 mM NaCl, and 5% glycerol, MMUT was further purified by ion exchange chromatography on a 2.5 × 10 cm Source Q column (Omnifit). MMUT was eluted with a linear gradient ranging from 0–70% Buffer 2 (50 mM HEPES pH 7.3, 500 mM NaCl, 5% glycerol). Fractions containing MMUT were dialyzed overnight against Buffer A (50 mM HEPES pH 7.5, 150 mM KCl, 2 mM $MgCl_2$, 2 mM TCEP, and 5% glycerol). Aliquots were flash-frozen and stored at −80 °C. Proteins were obtained in yields of 15 mg (wild-type), 7 mg (R228Q), 9 mg (R228Q/R616C), 10 mg (R616C), and 4 mg (R694W) per liter of culture.

### Expression and purification of MMAA
Wild-type and mutant MMAA were purified as reported[50]. The MMAA mutants R98G and R209S were generated by the Quickchange protocol using the following forward primers. The reverse primer had a complementary sequence.

R98G: 5′–CCGGTCTGATTCAAGGTCAAGGTGCGTGCCTG-3′

R209S:  5′–GTGACATGAACGCGTACATTAGTCCGAGCCCCGACC CG-3′

Wild-type and mutant MMAA were expressed in *E. coli* BL21 (DE3) in Terrific Broth at 37 °C until $OD_{600nm}$ reached 1.0–1.2. Then, 500 μM IPTG was added and incubated at 37 °C for 18–20 h. Cells were harvested by centrifugation at $5500 \times g$ and the cell pellet was stored at −80 °C until further use. The cell pellet was resuspended in lysis buffer in the presence of 100 μM phenylmethylsulfonyl fluoride and 1 tablet EDTA-free cOmplete™ Protease Inhibitor Cocktail (Roche). The suspension was sonicated, and cell debris was removed by centrifugation at $38,500 \times g$. The supernatant was loaded onto a Ni-NTA column, and the column was washed with 150 mL lysis buffer. MMAA was eluted from the column with a liner imidazole gradient (20–200 mM). Fractions containing MMAA were pooled and concentrated to 20–30 mL using an Amicon 30 kDa centrifugal device (EMD Millipore). His-tagged TEV protease was added (0.02 mg/mg of protein) and dialyzed overnight against 50 mM Tris, 300 mM NaCl, 1 mM DTT, 0.5 mM EDTA and 5% glycerol pH 8.0. The TEV-treated solution was loaded onto a Ni-NTA column and the cleaved MMAA, present in the flowthrough, was concentrated down to ~5 mL using an Amicon 30 kDa centrifugal device. MMAA was further purified by size exclusion chromatography on a 1.6 × 60 cm Hiload Superdex 200 column (GE Healthcare) pre-equilibrated with Buffer A. Fractions containing MMAA were concentrated, and flash frozen in liquid nitrogen.

### Expression and purification of MMAB
Wild-type MMAB were purified as reported[50]. Briefly, MMAB was expressed in *E. coli* BL21 (DE3) in Luria Bertani medium at 37 °C until the $OD_{600}$ nm reached ~0.6. The temperature was reduced to 20 °C and after 40 min, 50 μM IPTG was added. After 14–16 h incubation at 20 °C, cells were harvested by centrifugation at $38,500 \times g$ and the cell pellet was stored at −80 °C until further use. The cell pellet was resuspended in lysis buffer (50 mM Tris-HCl pH 8.0, 300 mM NaCl, 10 mM imidazole, and 1 mM TCEP) containing 100 μM phenylmethylsulfonyl fluoride and 1 tablet EDTA-free cOmplete™ Protease Inhibitor Cocktail (Roche). The suspension was sonicated, and cell debris was removed by centrifugation at 38,500 x *g*. The supernatant was loaded onto a Ni-NTA column and the column was washed with 150 mL lysis buffer. MMAB was eluted from the column with a liner imidazole gradient (20–300 mM). Fractions containing MMAB were pooled and concentrated to 20–30 mL using Amicon 30 kDa centrifugal device (EMD Millipore). Thrombin was added (5 units/mg of protein) and dialyzed overnight against 50 mM Tris, 300 mM NaCl, 1 mM TCEP 5% glycerol pH 8.0. Then, the solution was loaded onto a Ni-NTA column and the cleaved MMAB, present in the flowthrough, was concentrated down to ~20 mL, using an Amicon 30 kDa centrifugal device. The solution was then applied to a benzamidine column

(GE Healthcare) to remove thrombin. MMAB, present in the flow-through, was concentrated to ~5 mL and dialyzed overnight against Buffer A. Aliquots of MMAB were flash-frozen in liquid nitrogen and stored at −80 °C.

### AdoCbl loading to MMUT from MMAB

In a quartz cuvette, 15 μM AdoCbl and 15 μM MMAB in Buffer A (50 mM HEPES pH 7.5, 150 mM KCl, 2 mM MgCl$_2$, 2 mM TCEP and 5% glycerol) were incubated for 5 min at 25 °C to prepare MMAB•AdoCbl. AdoCbl transfer was initiated by the addition of a premixed solution of 15 μM MMUT 30 μM MMAA, 1 mM GMPPCP to the reaction mixture at 25 °C. The spectrum of AdoCbl was monitored between 300 and 700 nm and recorded every 60 sec for 30 min at 25 °C. Transfer of AdoCbl from MMAB to MMUT is signaled by an increase in absorbance at 525 nm.

### Cob(II)alamin off-loading from MMUT to MMAB

Cob(II)alamin off-loading from MMUT was monitored under anaerobic conditions (Omni-Lab anaerobic chamber, containing <0.5 ppm O$_2$) in a quartz cuvette containing Buffer A, 15 μM cob(II)alamin and 10 μM MMUT, incubated for 10 min at 20 °C. The "repair mixture" was prepared by mixing 5 mM ATP, 15 μM of MMAB, 30 μM MMAA and 1 mM GTP. Cob(II)alamin transfer was initiated by adding the repair mixture to the cuvette at 20 °C. The spectrum was recorded between 300 and 750 nm every 15 sec for 15 min. Transfer of cob(II)alamin from MMUT to MMAB is signaled by an increase in absorbance at 464 nm, which corresponds to 4-coordinate cob(II)alamin bound to MMAB.

### MMUT GAP activation of MMAA

To measure the GAP activation by MMUT variants, samples were prepared by mixing 2.5 μM MMAA and 2.5 μM wild-type or mutant MMUT in 50 mM HEPES pH 7.5, 150 mM KCl, 2 mM MgCl$_2$, 2 mM DTT and 5% glycerol in a 1.5 mL Eppendorf tube, and pre-incubated at 30 °C for 10 min. The intrinsic GTPase activity of wild-type, R98G and R209S MMAA GTPase was measured using 12.5 μM of each protein. The reaction was initiated by adding 3 mM GTP to the Eppendorf tubes at 30 °C. Aliquots of 50 μL were removed after 10 min and 20 min and the reaction was immediately terminated by adding 2.5 μL of 2 M trichloroacetic acid and placed on ice. The precipitant was removed by centrifugation at 15,870 x $g$ for 10 min at 4 °C and the GDP concentration in the supernatant was analyzed by HPLC as described previously[50]. The concentration of GDP present in the control sample (lacking proteins) was subtracted from each value. A calibration curve for GDP (0–500 μM) was obtained by treating standard samples as mentioned above. HPLC data were collected using Agilent Chemstation B.03.01software.

### Catalytic activity of MMUT

MMUT activity was assessed in the thiokinase-coupled spectrophotometric assay as described previously[51]. A stock solution of holo MMUT was prepared by mixing 10 μM MMUT with 20 μM AdoCbl in 100 mM potassium phosphate, 3 mM MgCl$_2$ pH 7.5 buffer followed by incubation at 30 °C for 15 min. In a quartz cuvette, 5 μM AdoCbl, 3 mM GDP, 5 μM thiokinase, 0.5 mM M-CoA, and 70 μM 5,5′-dithiobis(2-nitrobenzoic acid) was incubated at 30 °C in 100 mM potassium phosphate, 3 mM MgCl$_2$ pH 7.5 buffer in a final volume of 200 μL and the background thioesterase activity was recorded at 412 nm for 5 min. The reaction was initiated by the addition of 2 μL of holo-MMUT stock (2 nM final concentration) to the cuvette. The specific activity was calculated assuming that the amount of CoA detected was directly proportional to the concentration of succinyl-CoA formed by MMUT. An extinction coefficient of 14,150 cm$^{-1}$ M$^{-1}$ was used for the TNB$^-$ anion[52]. UV-visible data were collected on LabSolutionsUVVis for Shimadzu UV-1900i or UV-2600 software and analyzed in OriginPro 2022 9.9.0.225.

### Crystallography

The M$_2$C$_2$ complex with GDP and CoA was prepared by mixing 100 μM MMUT, 200 μM MMAA, 2 mM GDP, and 2 mM CoA in 50 mM HEPES, pH 7.5, 75 mM KCl, 2 mM MgCl$_2$, 2 mM TCEP. The mixture was purified on a Sephacryl S300 16/600 column (Cytiva) pre-equilibrated with the same buffer (Supplementary Fig. 9). Fractions (2.5 mL) were collected in tubes containing 2.5 μL 100 mM GDP and 2.5 μL 100 mM CoA (final concentration 100 μM each). The M$_2$C$_2$ complex-containing fractions were pooled and concentrated to ~30 mg/mL. GDP and CoA were added to a final concentration of 500 μM each. The complex was flash-frozen in liquid nitrogen and stored at −80 °C. Crystals of M$_2$C$_2$ appeared within a day using the hanging drop vapor diffusion method. Crystals with the best diffraction were obtained in a 1 μL drop containing 1:1 protein (15 mg/mL): well solution (Morpheus 1 F11 (Molecular Dimensions): 120 mM monosaccharides mix, 100 mM buffer system 3, pH 8.5, 30% precipitant mix 3)[53]. Crystals were harvested after a week and flash-cooled in liquid nitrogen for data collection. No additional cryoprotectant was used.

### Data collection, processing, and refinement

Diffraction data were collected at GMCA (23-ID-B) at Argonne National Laboratory using the JBluIce Graphical User Interface software. X-ray diffraction data were processed with autoPROC, using the default pipeline which includes XDS, Truncate, Aimless, and STARANISO (Tickle et al. STARANISO, 2018, Global Phasing Ltd, UK)[54–56]. Analysis by STARANISO showed that diffraction data were anisotropic, with diffraction limits along the reciprocal directions of 2.79 Å along $0.967a^* − 0.253c^*$, 3.62 Å along $b^*$, and 3.15 Å along $−0.05 a^*+0.999c^*$. Automated resolution cutoff with local $I/\sigma(I) > 1.20$ of anisotropic corrected data by STARANISO resulted in a 2.79 Å resolution data set. Data processing statistics are shown in Supplementary Table 1.

The structure was solved by molecular replacement with Phaser[57], using previously solved structures of MMUT (PDB code 2XIQ) and MMAA (PDB code 2WWW). The M$_2$C$_2$ crystals belonged to the space group P 21 with 4 chains of MMUT and MMAA per asymmetric unit. Iterative rounds of model building and refinement were performed with COOT[58] and refined in Phenix[59] or Refmac[55]. Ligand restraints were generated in *eLBOW*[60]. The geometric quality of the model was assessed in *MolProbity*[61]. Models were analyzed in Pymol (Schrödinger, LLC), and the structure figures were generated using UCSF Chimera[62]. Model refinement statistics are shown in Supplementary Table 1.

### Reporting summary

Further information on research design is available in the Nature Portfolio Reporting Summary linked to this article.

## Data availability

All data are available in the manuscript or supplementary materials. The structure factors and coordinates for human MMUT•MMAA•CoA•GDP have been deposited in the Protein Data Bank (PDB) under PDB cod: 8GJU. The following published PDB structures were used in this study: 3BIC, 2XIJ, 2XIQ, 2WWW, 2QM7, 8DPB, 4XC8.

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

## Acknowledgements
R.M. thanks Dr. Dali Liu (Loyola University Chicago) for helpful discussions on structure refinement. This work was supported in part by grants from the National Institutes of Health (K99-GM1434820 to RM and RO1-DK45776 to R.B.). GM/CA at APS has been funded by the National Cancer Institute (ACB-12002) and the National Institute of General Medical Sciences (AGM-12006, P30GM138396). This research used resources from the Advanced Photon Source, a U.S. Department of Energy (DOE) Office of Science User Facility operated for the DOE Office of Science by Argonne National Laboratory under Contract No. DE-AC02-06CH11357. The Eiger 16 M detector at GM/CA-XSD was funded by NIH grant S10 OD012289.

## Author contributions
R.M. and M.Y. established crystallization conditions and R.M. elucidated the crystal structure. M.R., H.G., and N.H. designed and performed the biochemical analyses. All authors helped conceive the experiments and analyze data. R.M., M.R., and R.B. drafted the manuscript, and all co-authors edited it.

## Competing interests
The authors declare no competing interests.
