## [Peer Review File · Nature Communications]

REVIEWERS' COMMENTS

Reviewer #1 (Remarks to the Author):

Mascarenhas et al. report the structure of the quaternary complex of human methylmalonyl-CoA mutase (MMUT), the G-protein chaperone MMAA, CoA, and GDP/Mg²⁺. The present 2.8 Å resolution structure of the ring-shaped complex, which has already shown in their previous paper (Cell Chem. Biol., 26, (2019) 960-969) at very low resolution, reveals detailed interactions between MMUT and MMAA. Also, the structure displays a remarkable structural feature that MMAA wedges between the B₁₂ and substrate domains of MMUT, which enables to unload the inactivated cofactor (cob(II)alamin). The manuscript is generally well written and easy to follow, but I have some concerns that need to be addressed before publication.

In the previous paper they showed that the mixture of MMUT and MMAA contained M₂C₁ complex, MMUT, and MMAA, rather than the annular M₂C₂ complex in the presence of GDP without CoA. Addition of CoA to the mixture drives the formation of the M₂C₂ complex?

Figure 3A. I cannot see the surface model of the interconnecting belt in the M₂C₂ complex. Is this missing due to disorder?

The color codes for Chain A and B of MMAA are dark and light shaded in Figure 2, while they are reversed in Figure 4A. This is confusing.

Supporting Information p.2, l. 6: 5'-deoxyadenosine should be 5'-deoxyadenosine.

Reviewer #2 (Remarks to the Author):

This paper addresses the molecular mechanism of vitamin B12 repair for methylmalonyl-CoA mutase (MMUT) by the proteins MMAA and MMAB. MMAB is an adenosyltransferase that converts cobalamin to 5'-deoxyadenosylcobalamin while MMAA is a GTPase that facilitates removal of cobalamin from MMUT. While structures of bacterial orthologs of MMUT and MMAA, including *M. extorquens* MeaB (MMAA) and MutAB (MMUT) and *C. metallidurans* IcmF, are available, there are significant differences between these systems and the human proteins. Because mutations in the human proteins lead to methylmalonic aciduria, it is important to understand the process of cobalamin offloading to MMAA.

The authors have determined the crystal structure of a human MMUT/MMAA complex in the presence of CoA and GDP. A drastic conformational change of the MMUT B12 domain, stabilized by MMAA, exposes the B12 binding site and establishes new protein-protein interfaces between MMUT and MMAA. MMAA also exhibits major conformational changes upon binding MMUT, altering the GDP location and resolving the switch loops. The physiological relevance of the structure is demonstrated by reduced GTPase activating protein activity of MMUT in variants with MMAA/MMUT interface mutations. In the proposed functional model, the structure is suggested to be a "holding structure" primed for loading of AdoCbl upon GDP exchange for GTP. The findings not only explain disease-related mutations, but have implications for B12 loading mechanisms of other systems. Given that this study significantly advances understanding of this system, it is appropriate for publication in Nat. Commun.

Specific comments:

B12 should be introduced more explicitly at the beginning. A figure showing the structures of cobalamin and AdoCbl should be included, either as part of Fig. 1 or in the SI. The introduction does not make clear the distinctions among the key species, and it is not clear that AdoCbl is B12. Line 36 defines B12 as cobalamin.

Fig. 2C,D caption: please specify that CoA is green, unclear what "50" means

Fig. 3A: I don't see the dark grey and light grey structures here. It may be helpful to have a side-by-side comparison of the structures as well. The mix of colors and the inclusion of the surface is a lot to take in and distracts from the huge conformational change.

Fig. 3D caption: "alos" should be "also"

lines 193 and 214: unclear why these two sections have the same subheading.

Reviewer #3 (Remarks to the Author):

Summary

Cobalamin (B12) is a crucial cofactor in the enzymatic processes of the body, specifically in the two mammalian enzymes, methionine synthase and mitochondrial methylmalonyl-CoA mutase (MMUT). These enzymes use cobalamin to cycle between different oxidation states, but the cobalt ion in cobalamin is susceptible to inactivation. Certain proteins, such as MMAA and MMAB, are involved in maintaining, repairing, and reactivating the cobalamin. They facilitate a complex set of interactions, converting inactive cob(II)alamin into an active form, and then reloading it onto the enzyme. Mutations in these mitochondrial B12 trafficking proteins can cause hereditary methylmalonic aciduria, a metabolic disorder.

There is limited understanding of the structural basis for how inactive human MMUT signals to and recruits MMAA and MMAB for cofactor loading/off-loading. Therefore, the authors report the crystal structure of a complex involving human MMUT and MMAA, which reveals substantial conformational changes. These changes involve a 180° rotation of the B12 domain in relation to the substrate domain in MMUT, which is stabilized by MMAA. The authors claim that this rotation, facilitated by a variety of complex interactions, primes the B12 domain for cofactor loading and off-loading; this may provide insights into the mechanistic basis of pathogenic mutations that occur at the MMUT-MMAA interfaces.

Overall evaluation

The authors present a thorough structural characterization of the MMUT:MMAA-complex and the interfaces formed in the complex. Their results are mainly focused on describing the protein-protein and protein-cofactor/ligand interactions. However, the authors also derive hypotheses on the relevance of the observed interactions with respect to pathogenic mutations that have been reported to be involved in the formation of the disease methylmalonic aciduria. Using enzymatic analyses, the relevance of these mutations for the disease context are demonstrated, further strengthening the validity of the observed complex crystal structure.

Overall, the study is carried out with great expertise and the findings appear to be experimentally validated. My main point of criticism is, however, that the findings are rather limited in their scope:

After all, the authors present only a complex crystal structure in combination with a mutational analyses that verify the structural conclusions. Yet, this statement does not diminish the good quality of the provided data.

Minor points

- The authors use the term nanomotor when referring to MMAA. However, I do not regard MMAA as a nanomotor. Overall, nanomotors in biology refer to miniature machines capable of converting energy into directed motion at the nanoscale with the ability to exert the motion repetitively. GTPases such as MMAA rather use the energy provided by GTP to stabilize an energetically unfavorable state in which multiple GTP-turnover is commonly not involve. Therefore, in my opinion, the term nanomotor is incorrectly used in the context of MMAA.

- author statement „The B12 is rotated $\sim 180^\circ$ in the M2C2 complex versus in MMUT, which is evident from overlays of MMUT in the complex on apo-MMUT (PDB code 3BIC), MMUT•AdoCbl (PDB code 2XIJ) or MMUT•CoA•AdoCbl (PDB: 2XIQ) ($C\alpha$ RMSD across all pairs= 17 \AA) (Figure 3A, Movie S1).”: Do the authors mean “The B12 domain is rotated $\sim 180^\circ$...?”

Please also check for other occurrences of this ambiguity in the text.

- author statement “The intrinsic GTPase activity of MMAA ($0.06 \pm 0.02 \text{ min}^{-1}$), which increases 36-fold ($2.1 \pm 0.2 \text{ min}^{-1}$) in the presence of wild-type MMUT, is attenuated 7- (R616C) and 12-fold (R228Q and R694W) in the variants (Table S2).”: Please correct to “...is attenuated to ...”.

- author statement “The GTPase activating protein (GAP) activity of MMUT serves as a proxy for its complex formation with MMAA.”: Please provide a reference that validates this statement.

- discussion, author statement “Patient mutations at the MMAA-MMUT interfaces impact complex formation...”: The authors have not shown directly that complex formation is affect by the patient mutations. Rather, they observed a decrease in catalytic efficiencies for the mutated proteins. Whether this is due to a weakened complex formation, has not been shown (though likely). Therefore, please rephrase accordingly.

- The structural videos can be improved. In doing so, the morphing videos should cycle between the beginning and end states without abrupt jumps from the end to the starting state. This facilitates the

understanding of the structural dynamics. Furthermore, the domains should be labeled or shown in different colors so that the uninformed reader can recognize the individual parts of the structure.

Reviewer #1 (Remarks to the Author):

1) *In the previous paper they showed that the mixture of MMUT and MMAA contained M_2C_1 complex, MMUT, and MMAA, rather than the annular M_2C_2 complex in the presence of GDP without CoA. Addition of CoA to the mixture drives the formation of the M_2C_2 complex?*

Addition of CoA stabilized the annular M_2C_2 complex, and this stoichiometry was revealed by the crystal structure. In our previous study we assigned the stoichiometry of the complexes by size exclusion chromatography and multi-angle light scattering. However, the structure revealed that the analytical gel filtration analysis was not accurate enough to distinguish between M_2C_2 (480 kDa) and M_2C_1 (400 kDa).

2) *Figure 3A. I cannot see the surface model of the interconnecting belt in the M_2C_2 complex. Is this missing due to disorder?*

Yes, the interconnecting belt is flexible and only partially resolved in the M_2C_2 structure as stated in the manuscript: “. The swinging out of the B_{12} domain is enabled by a flexible interconnecting belt (~100 residues), which is partially unresolved in the M_2C_2 structure (between 580-595), suggesting mobility.”

3) *The color codes for Chain A and B of MMAA are dark and light shaded in Figure 2, while they are reversed in Figure 4A. This is confusing.*

We thank the reviewer for pointing this out and have changed the colors in Fig. 4A to match Fig. 2.

4) *Supporting Information p.2, l. 6: 5'-deoxyadenosine should be 5'-deoxyadenosine.*

Corrected

Reviewer #2 (Remarks to the Author):

1) *B_{12} should be introduced more explicitly at the beginning. A figure showing the structures of cobalamin and AdoCbl should be included, either as part of Fig. 1 or in the SI. The introduction does not make clear the distinctions among the key species, and it is not clear that AdoCbl is B_{12} . Line 36 defines B_{12} as cobalamin.*

Fig. 1A includes cartoons of B_{12} and serves as an introduction to the cofactor for the general audience. With due respect, we believe that it is unnecessary to include chemical structures of cobalamin derivatives in a study that focuses on the apo-structures of a trafficking intermediate. The term cobalamin is in fact broadly used to describe B_{12} derivatives. We have specified that 5'-deoxyadenosyl cobalamin (AdoCbl) is the active form of the cofactor for MMUT on line 42.

2) *Fig. 2C,D caption: please specify that CoA is green, unclear what "50" means*

We thank the reviewer for pointing this out and have corrected the figure legend.

3) *Fig. 3A: I don't see the dark grey and light grey structures here. It may be helpful to have a side-by-side comparison of the structures as well. The mix of colors and the inclusion of the surface is a lot to take in and distracts from the huge conformational change.*

We have edited Figure 3A as suggested. The interconnecting belt is now shown as a ribbons and we have changed the shades of grey to be more distinctive.

4) Fig. 3D caption: "alos" should be "also"

Corrected

5) lines 193 and 214: unclear why these two sections have the same subheading.

Line 193 is labelled as MMUT patient mutations disrupt complex formation and Line 214 is labelled as MMAA patient mutations disrupt complex formation.

Reviewer #3 (Remarks to the Author):

1) The authors use the term nanomotor when referring to MMAA. However, I do not regard MMAA as a nanomotor. Overall, nanomotors in biology refer to miniature machines capable of converting energy into directed motion at the nanoscale with the ability to exert the motion repetitively. GTPases such as MMAA rather use the energy provided by GTP to stabilize an energetically unfavorable state in which multiple GTP-turnover is commonly not involve. Therefore, in my opinion, the term nanomotor is incorrectly used in the context of MMAA.

MMAA fits the reviewer's own definition that nanomotors are capable of converting energy into directed motion on a nanoscale. We disagree with the suggestion that MMAA is not catalytic (we have reported its k_{cat}) and presumption about the stoichiometry of GTP hydrolyzed to stabilize the M_2C_2 complex. For these reasons, we have chosen to retain use of the term.

2) author statement „The B12 is rotated $\sim 180^\circ$ in the M_2C_2 complex versus in MMUT, which is evident from overlays of MMUT in the complex on apo-MMUT (PDB code 3BIC), MMUT•AdoCbl (PDB code 2XIJ) or MMUT•CoA•AdoCbl (PDB: 2XIQ) ($C\alpha$ RMSD across all pairs=17 Å) (Figure 3A, Movie S1).”: Do the authors mean “The B12 domain is rotated $\sim 180^\circ$...?”

Please also check for other occurrences of this ambiguity in the text.

We thank the reviewer for pointing this out and have corrected the statement.

3) author statement “The intrinsic GTPase activity of MMAA ($0.06 \pm 0.02 \text{ min}^{-1}$), which increases 36-fold ($2.1 \pm 0.2 \text{ min}^{-1}$) in the presence of wild-type MMUT, is attenuated 7- (R616C) and 12-fold (R228Q and R694W) in the variants (Table S2).”: Please correct to “...is attenuated to ...”.

We believe the statement is appropriate as is.

4) author statement “The GTPase activating protein (GAP) activity of MMUT serves as a proxy for its complex formation with MMAA.”: Please provide a reference that validates this statement.

We have added a reference (Cell Chem. Biol., 26, (2019) 960-969).

5) discussion, author statement “Patient mutations at the MMAA-MMUT interfaces impact complex formation...”: The authors have not shown directly that complex formation is affect by the patient mutations. Rather, they observed a decrease in catalytic efficiencies for the mutated proteins. Whether this is due to a weakened complex formation, has not been shown (though likely). Therefore, please rephrase accordingly.

Since the GAP activity of MMUT, and cofactor transfer between MMAB and MMUT, both require complex formation (between MMUT and MMAA), our statement is appropriate. Furthermore, in a previous study (Cell Chem. Biol., 26, (2019) 960-969), we had demonstrated that MMAA patient mutations impact complex formation.

6)The structural videos can be improved. In doing so, the morphing videos should cycle between the beginning and end states without abrupt jumps from the end to the starting state. This facilitates the understanding of the structural dynamics. Furthermore, the domains should

be labeled or shown in different colors so that the uninformed reader can recognize the individual parts of the structure.

We have incorporated these suggestions.